# Ambulatory specialist costs and morbidity of coordinated and uncoordinated patients before and after abolition of copayment: A cohort analysis

**Michaela Olm**[1]*, **Ewan Donnachie**[2], **Martin Tauscher**[2], **Roman Gerlach**[2], **Klaus Linde**[1], **Werner Maier**[3], **Lars Schwettmann**[3,4], **Antonius Schneider**[1]

**1** TUM School of Medicine, Institute of General Practice and Health Services Research, Technical University of Munich, Munich, Bavaria, Germany, **2** Bavarian Association of Statutory Health Insurance Physicians, Munich, Bavaria, Germany, **3** Institute of Health Economics and Health Care Management, Helmholtz Zentrum München, German Research Center for Environmental Health (GmbH), Neuherberg, Bavaria, Germany, **4** Department of Economics, Martin Luther University Halle-Wittenberg, Halle an der Saale, Saxony-Anhalt, Germany

* michaela.olm@mri.tum.de

**Data Availability Statement:** The data that support the findings of this study are held by the Bavarian Association of Statutory Health Insurance

## Abstract

To strengthen the coordinating function of general practitioners (GPs) in the German health-care system, a copayment of €10 was introduced in 2004. Due to a perceived lack of efficacy and a high administrative burden, it was abolished in 2012. The present cohort study investigates characteristics and differences of GP-coordinated and uncoordinated patients in Bavaria, Germany, concerning morbidity and ambulatory specialist costs and whether these differences have changed after the abolition of the copayment. We performed a retrospective routine data analysis, using claims data of the Bavarian Association of the Statutory Health Insurance Physicians during the period 2011–2012 (with copayment) and 2013–2016 (without copayment), covering 24 quarters. Coordinated care was defined as specialist contact only with referral. Multinomial regression modelling, including inverse probability of treatment weighting, was used for the cohort analysis of 500 000 randomly selected patients. Longitudinal regression models were calculated for cost estimation. Coordination of care decreased substantially after the abolition of the copayment, accompanied by increasing proportions of patients with chronic and mental diseases in the uncoordinated group, and a corresponding decrease in the coordinated group. In the presence of the copayment, uncoordinated patients had €21.78 higher specialist costs than coordinated patients, increasing to €24.94 after its abolition. The results indicate that patients incur higher healthcare costs for specialist ambulatory care when their care is uncoordinated. This effect slightly increased after abolition of the copayment. Beyond that, the abolition of the copayment led to a substantial reduction in primary care coordination, particularly affecting vulnerable patients. Therefore, coordination of care in the ambulatory setting should be strengthened.

Physicians. The data are not publicly available due to data protection regulations, but may be made available to researchers upon reasonable request and with the permission of the Bavarian Association of Statutory Health Insurance Physicians (contact via versorgungsforschung@kvb.de).

**Funding:** The study was funded by the Central Research Institute for Ambulatory Health Care in Germany (Zentralinstitut für die Kassenärztliche Versorgung in Deutschland, https://www.zi.de/). MO's position was financed from the funds. The funders had no role in study design, data collection and analysis, decision to publish, or preparation of the manuscript.

**Competing interests:** I have read the journal's policy and the authors of this manuscript have the following competing interests: ED, MT and RG are employees of the Bavarian Association of Statutory Health Insurance Physicians. This does not alter our adherence to PLOS ONE policies on sharing data and materials. The authors MO, KL, WM, LS and AS declare that they have no competing interests.

# Introduction

A preponderant issue affecting nearly all developed countries is an ageing society, with higher levels of chronic diseases and multimorbidity [1] and increasing healthcare expenditures [2]. At the same time, efforts have been made to facilitate healthcare access, especially for people with high vulnerability [3]. This, coupled with more expensive treatment options, creates conflict over allocating scarce resources. Unnecessary treatments and multiple diagnostic procedures, for example as a result of multiple specialist utilisation, often referred to as 'doctor shopping', exacerbate this situation [4].

Effective and needs-oriented care is therefore essential. Past studies have demonstrated that comprehensive primary care can contribute decisively to an effective care, and that the coordinating role of the primary care physician is crucial [5–8]. Based on a trusting doctor-patient relationship, a high level of continuity of care can be achieved [9,10]. In addition, comprehensive primary care has the potential to support free access to health services [5], benefitting vulnerable [9] and deprived groups [5].

In Germany, ambulatory care is characterised by a number of distinctive features. Some internists without specialisation are licensed as family physicians, with both, family physicians and internists in family practice considered to be general practitioners (GPs). Another characteristic is the fact that specialist physicians also participate in the primary care system, mainly comprising dermatologists, ear, nose and throat (ENT) specialists, gynaecologists, internists with and without specialisation (e.g. cardiology, gastroenterology, pulmonology and oncology), neurologists, ophthalmologists, orthopaedics, psychiatrists, psychotherapists (both physician and non-physician), radiologists, surgeons, and urologists.

Compared to other countries, such as the United Kingdom (UK), Germany has a weak primary care system in terms of coordination [11], as patients have free and direct access to all licensed GPs and primary care specialists. Germany has a very high physician contact rate with 14.7 physician contacts per patient annually in 2016 [12]. In 2004, a copayment was introduced to reduce this high number and to strengthen the coordinating function of GPs. For each physician contact within a three-month billing period, a fee of €10 had to be paid unless a referral was made from another physician [13]. However, as the influence on physician contacts was considered to be too low in relation to the high bureaucracy required, the copayment was abolished in 2012 [14].

A study analysing data from the year 2011 found that patients consulting a specialist with a GP referral, generated (on average) €9.65 lower healthcare expenditures than patients who directly consulted a specialist [15]. Other studies assessing the impact of the German copayment were performed after its introduction [16–18] or before its abolition [19–21]. Assessing before and after in the same study will increase our understanding of the impact of care coordination on resource use in the German ambulatory healthcare system and how coordination of care can be affected in different patient groups. Previous time series analyses found a large decrease of coordination of care on a population-based level [22]. The aim of our cohort analysis is to analyse characteristics and differences between GP-coordinated and uncoordinated patients concerning morbidity and ambulatory specialist costs. Furthermore, we investigated whether these differences have changed after the copayment abolition.

# Materials and methods

## Study design

A retrospective routine data analysis was performed using ambulatory claims data from Bavaria, Germany. A cohort was observed between the years 2011 and 2016, encompassing 2

years with copayment and 4 years following its abolition in January 2013. Due to limitations in the data quality, with respect the present analysis, time periods before 2011 were not considered. We included all statutorily insured patients aged ≥18 years in 2011 who were resident in Bavaria. Patients were excluded if not observed following the abolition of the copayment in 2013. For computational reasons, a random sample of 500 000 patients was selected from all patients meeting these criteria. This closed cohort is not therefore influenced by the addition of new patients, for example migrants or those reaching the age of 18. However, patients could exit the cohort for example due to death, residency outside of Bavaria, or a switch to private health insurance.

## Population and sources of data

Bavaria, situated in the south of Germany, is the largest federal state by area, with a population of 13 million people living in a mix of large cities and rural areas [23]. The underlying database is held by the Bavarian Association of Statutory Health Insurance Physicians (German: *Kassenärztliche Vereinigung Bayerns*, KVB) and covers all statutorily insured outpatients in Bavaria, corresponding to ~85% of the Bavarian population.

For remuneration purposes, patient-level data are submitted by GPs (~9 000), specialists in outpatient care (~13 000) and psychotherapists (~4 000). The submitted data contain the items billed according to a standardised fee schedule along with the corresponding medical diagnoses, coded by the German modified ICD-10 classification (International Classification of Diseases, 10th Revision). In Germany, billing occurs on a quarterly basis, generally with bundled payment for each treatment episode, i.e. a patient treated in a given quarter by a single outpatient practice. Additional fee-for-service payments are made for technical or time-consuming services. In primary care, this includes lung function testing, ultrasound, or chronic disease management, whereas in specialist care, most services are provided on a fee-of-service basis. Deductions are made if a quarterly practice budget is exceeded, thus discouraging overuse.

## Definition of coordinated care

A patient was classified as 'coordinated' if every regular specialist visit within a quarter occurred as a result of a GP referral (coordinated patient, **CP**) [15,20,22]. In contrast, a patient who consulted at least one specialist within a quarter without a referral was defined as uncoordinated (uncoordinated patient, **UP**). When determining the coordination status, specialist contacts were considered if a referral from a GP would be expected in the context of coordinated care. In particular, emergency treatment, pregnancy care or routine screening (e.g. mammography) was excluded from consideration. Similarly, consultations with radiologists, anaesthetists, surgeons, nuclear physicians and dialysis centres often occur on referral from a specialist and were therefore deemed not relevant when determining the GP coordination status (not relevant for coordinated care, **NR**). Patients consulting only a GP within a quarterly period were categorised as 'GP care only' (**GP**).

## Patient characteristics

Analysed patient characteristics were regional differences, differences in morbidity and ambulatory specialist costs.

To assess regional differences, we have used the Bavarian Index of Multiple Deprivation 2010 (BIMD 2010) at the district level to account for regional differences [24]. Adapting the UK model [25] to the German context, the BIMD 2010 combines official socio-demographic, socio-economic and environmental data, divided into seven domains of deprivation: income, employment, education, municipal or district revenue, social capital, environment and

security. The BIMD 2010 is divided into quintiles, with the 1st quintile (Q1) representing the lowest deprivation and the 5th quintile (Q5) representing the highest.

Differences between large towns and rural areas were assessed using the settlement structure classification of the Federal Institute for Research on Building, Urban Affairs, and Spatial Development (German: *Bundesinstitut für Bau-, Stadt- und Raumforschung*, BBSR) [26]. The four groups are 'large cities' (i.e. more than 100 000 inhabitants), 'urban districts', 'rural districts showing densification' and 'sparsely populated rural districts'.

Morbidity was assessed using the KM87a_2015 grouper, which specifies 72 aggregated medical condition categories (Top Hierarchical Condition Categories (THCC), and Rest Hierarchical Condition Categories, (RHCC)) and represents a convenient, cost-based tool for analysing complex ICD-10 diagnoses [27]. The grouper was developed in the United States and modified for the German healthcare system by the Institute for Strategic Assessment of Reimbursement for Medical Services (German: *Institut des Bewertungsausschusses*, InBA), an official organ of the German Ministry of Health. Separately, patients with mental disorders were identified by relevant diagnosis groups (THCC054, THCC055, THCC057, THCC058, THCC060), largely corresponding to the documentation of the ICD-10 F-Code. Chronic diseases were extracted using an official InBA-list, which condenses the most important chronic diseases [28].

Cost differences between coordinated and uncoordinated patients were estimated using ambulatory specialist costs. These costs represent the amount in Euros subsequent to deductions due to budgetary constraints and other billing regulations. The copayment is not included, as attending physicians only collected the €10 on behalf of the statutory health insurances. Of interest is the effect of GP-centred coordination of care on the cost of specialist care, both with and without the presence of the copayment. The cost of GP care cannot be reliably investigated because up to 1 million predominantly older patients were enrolled in a separate care model for which limited GP claims data is available (German: *Hausarztzentrierte Versorgung*, HzV).

## Statistical procedures

The health care utilisation of each member of the cohort was summarised for each quarterly billing period during the observation period. Cohort members without consultation in a given quarter were recorded with zero utilisation, with coordination status carried over from the previous quarter. The characteristics 'age', 'sex' and district of residence are as recorded in 2011 and remained unchanged during the analysis. The effect of patient ageing is thus incorporated in the time variable (each quarter as an own factor).

A consultation of a single practice (GP or specialist) within a 3-month period (quarter) is defined as a treatment episode. If a patient consults the same practice for different reasons within a quarter, both contacts are merged for administrative purposes to form a single episode.

The longitudinal analysis was carried out in two stages. First, a descriptive analysis of the cohort was conducted in tabular and graphical form. In order to visualise changes following the copayment abolition, a time series analysis was performed. Measures of interest are the cohort size, mean age, gender distribution, proportion of patients living in cities, number of medical condition categories (THCCs and RHCCs), proportions of patients with chronic or mental disorders, number of cases per patient, and financial claims of GPs and specialists. These trends were also stratified by the four coordination groups as defined above: coordinated patient (CP), uncoordinated patient (UP), GP care only (GP), and not relevant for coordinated care (NR).

In a second stage, marginal structural models (MSM) were applied in order to estimate the causal effect of coordination before and after the abolition of the copayment. The MSM paradigm combines weighting with regression to create a doubly robust estimate of the causal effect of interest [29–31]. Inverse probability of treatment weights (IPTW) were calculated using multinomial regression models to determine the probability that a patient belonged to each of the four coordination groups [29,32]. In order to assess the importance of a single co-variable and guarantee the robustness of the results, the models were built up successively to investigate the impact of adding age, sex, regional and morbidity parameters (Table in S1 Table and Figure in S1 Fig). The distribution of probabilities between 2011 and 2016, and further explanations are presented in an additional file (Figure in S2 Fig).

The cost of specialist care was modelled using IPTW-weighted longitudinal regression models (generalised estimating equations, GEE) [33]. The effect of coordination before and after the abolition of the copayment was captured by adding the corresponding interaction term to the model. Again, model covariables were added successively in order to understand the behaviour of the model (Table in S2 Table).

## Data protection

The study was carried out following the German guideline 'Good Practice for Secondary Data Analysis' (German: *Gute Praxis Sekundärdaten*) [34]. According to this guideline, ethics approval and patient consent are not required for studies based solely on anonymised claims data. Nevertheless, an approval was obtained from the responsible data protection officer of the Bavarian Association of Statutory Health Insurance Physicians and the analyses were conducted under strict data protection restrictions.

## Results

### Descriptive analysis

Basic characteristics of the cohort are shown in Table 1, presenting the first quarter of 2012 (1/2012) under influence of the copayment. In addition, a Table in S3 Table presents the first quarter of 2014 (1/2014), following abolition of the copayment. The development over the entire period (2011–2016) is shown in Figs 1–3.

Table 1. Cohort characteristics at the time of the copayment (1st quarter 2012) (CP, UP, GP, NR).

|  | Total | CP | UP | GP | NR |
|---|---|---|---|---|---|
| Number of patients (%) | 502 542 | 114 867 (22.9) | 152 274 (30.3) | 174 148 (34.7) | 61 253 (12.2) |
| Age (mean) | 49.3 | 56.1 | 49.0 | 48.7 | 39.2 |
| Gender: female (%) | 56.2 | 58.0 | 55.8 | 47.2 | 79.4 |
| Proportion with residence 'city' (%) | 44.7 | 43.3 | 49.2 | 41.6 | 44.7 |
| Number of medical condition categories/patient (mean) | 5.6 | 8.2 | 5.8 | 4.3 | 4.0 |
| Number of cases/patient (mean) | 2.2 | 3.5 | 2.7 | 1.0 | 1.9 |
| Proportion with chronic diseases (%) | 57.2 | 80.3 | 54.0 | 52.0 | 37.0 |
| Proportion with mental diseases (%) | 28.1 | 40.5 | 30.2 | 20.4 | 21.2 |
| Specialist financial claims in € (mean) | 87.4 | 138.1 | 153.3 | 2.3 | 70.4 |
| General practitioner financial claims in € (mean) | 48.8 | 73.3 | 38.3 | 47.6 | 32.7 |

Note: Coordination categories: CP: Coordinated patient (specialist contact with referral); UP: Uncoordinated patient (specialist contact without referral); GP: General practitioner care only (no specialist contact); NR: Not relevant for coordinated care.

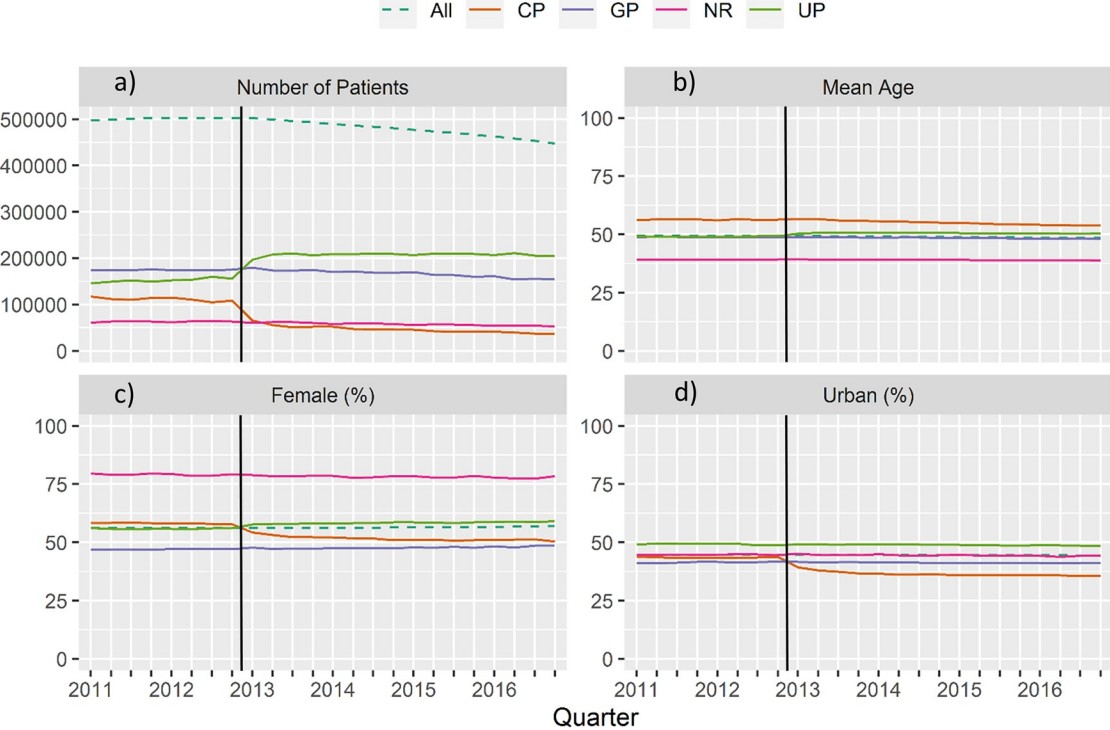

**Fig 1.** Cohort time series: (a) Number of patients, (b) mean age, (c) proportion with female gender (%), (d) and proportion of urban population (%), divided into coordination status coordinated patients (CP), uncoordinated patients (UP), GP only (GP), not relevant for coordinated care (NR), and total cohort, quarterly (vertical line marks the point of abolition).

In 2012, 22.9% of all patients were coordinated (CP) and 30.3% were uncoordinated (UP) (Table 1). In addition, CPs were on average older, more often women, had higher levels of morbidity (more medical condition categories, higher proportions of chronic or mental diseases), resided more often in rural areas and were associated with higher GP financial claims, but lower specialist costs. These differences are also visible in the GP and NR groups, with a higher proportion of women in the NR group.

**Cohort size.** Between 2011 and 2016, the number of cohort participants decreased by approximately 10% due to death, change of residence to outside Bavaria, or change to private health insurance (Fig 1a).

**Coordination-subgroups.** On abolition of the copayment on January 2013, substantial changes occurred in the groups coordinated patients (CP) and uncoordinated patients (UP) (Fig 1a). Fewer patients were coordinated and more were uncoordinated. For the remainder of the observation, the two groups remained largely stable. Additionally, the NR and GP showed a slight decrease in number. It should therefore be noted that the following descriptive results reflect the change in make-up of the results over time, due primarily to the abolition of the copayment.

**Age, sex & residence.** The mean age of the groups remained almost stable. After 2012, CP showed a small decrease, whereas the mean age of the UP group increased, also only slightly (Fig 1b). In contrast, the proportion of female CPs decreased substantially, while the proportion of uncoordinated female patients increased. GP only patients and the total cohort showed small increases. The NR group has a high female proportion (~80%), which decreased slightly over time (see Fig 1c). This is due to the definition of the NR group incorporating patients

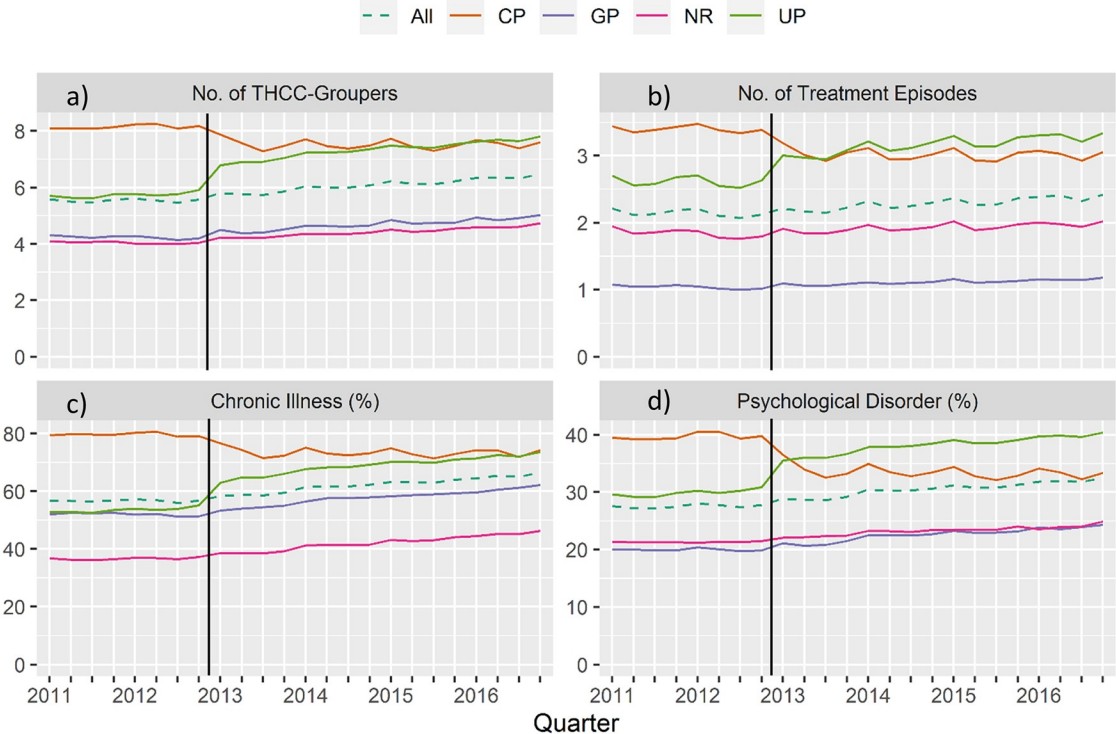

**Fig 2.** Cohort time series: (a) Number of medical condition categories, (b) number of cases/patient, (c) proportion with chronic diseases (%), (d) proportion with mental diseases (%), divided into coordination status coordinated patients (CP), uncoordinated patients (UP), GP only (GP), not relevant for coordinated care (NR) and total cohort, quarterly (vertical line marks the point of abolition).

with pregnancy care and mammography without additional physician contacts. For the proportion of patients with a 'city' residence (Fig 1d), only the CP group decreased, while the other categories remained stable.

**Morbidity.** Fig 2 summarizes the morbidity of the cohort over time, presenting the mean number of medical condition categories (THCC/RHCC), the mean number of treatment episodes per patients, and the proportions with chronic and mental diseases. While the number of medical condition categories per patient decreased in the CP group after 2012 (Fig 2a), it increased in all other categories, especially in the UP group. A similar trend was also visible for chronic (Fig 2c), and mental (Fig 2d) diseases. For the number of quarterly cases per patient (Fig 2b), a slight change in UP and CP was observable, while GP and NR remained nearly stable.

## Cost analysis

Fig 3 presents the GP and specialist costs over time. Overall, the mean financial claims of GPs (Fig 3a) and specialists (Fig 3b) increased. Again, the CP and UP groups exhibit substantial changes on abolition of the copayment. For the latter, specialist and GP costs increased, while costs for the former decreased. Additionally, claims for patients in the category 'not relevant for coordination' slightly increased.

Longitudinal regression modelling, estimating the mean specialist claim per patient (Table 2 and Figure in S3 Fig), indicates that an uncoordinated patient (UP) with certain characteristics (age, sex, morbidity and residency) had €21.78 higher specialist costs (mean

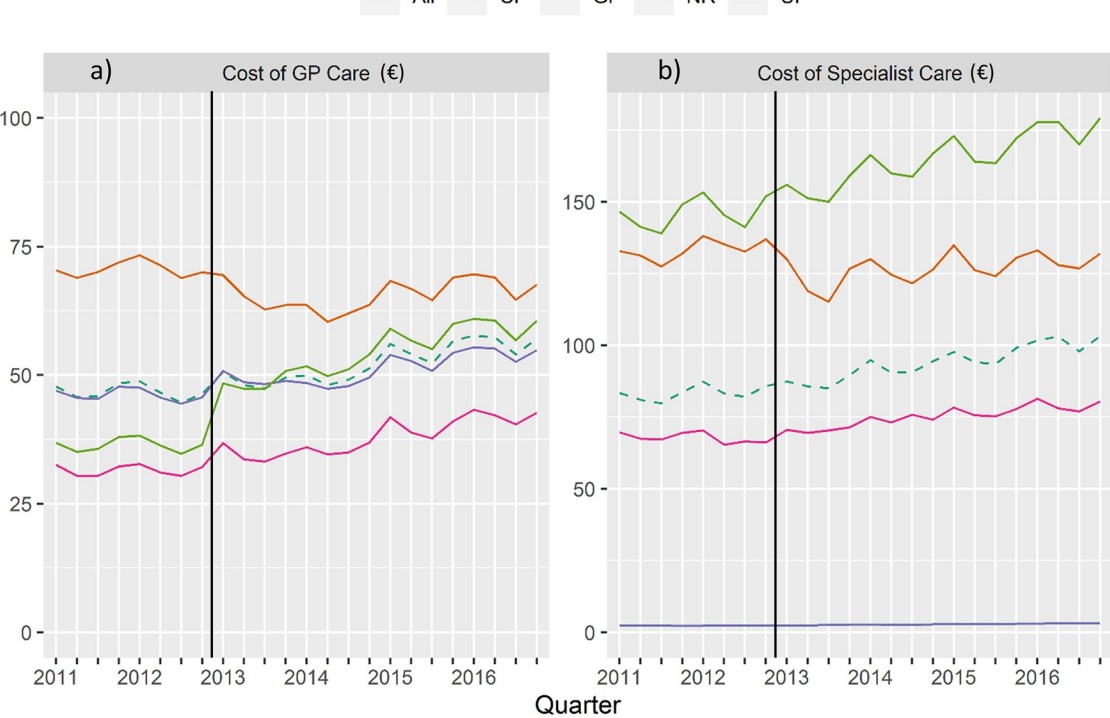

**Fig 3.** Cohort time series: (a) General practitioner financial claims in € and (b) specialist financial claims in € (means claimed per patient), divided into coordination status coordinated patients (CP), uncoordinated patients (UP), GP only (GP), not relevant for coordinated care (NR) and total cohort, quarterly (vertical line marks the point of abolition).

difference) than a coordinated patient (CP) with the same characteristics before abolition of the copayment. After abolition, the specialist costs of CP decreased by €5.55 and the specialist costs of UP decreased by €3.16. The latter effect is detected by interaction analysis of coordination and the presence of copayment (€5.55–€2.39). Thus, the difference between the CP and UP group after abolition was €24.94 (€21.78+€3.16).

Compared to the CP, the NR (−€50.82), and especially the GP (−€102.94) had substantially lower specialist costs. Again, this represents the difference to the reference category CP. The relatively high negative value in the GP group is because patients who only have contact with their GP in a quarter, have specialist costs of almost €0 on average (see also Fig 3b).

Female patients had higher specialist costs (difference male vs. female: €19.36) than males. As female patients are more often uncoordinated than male patients (Figure in S1 Fig), an interaction between age and sex was tested. The result indicated that specialist costs for women decreased with increasing age. In addition, patients with mental (€22.24) and chronic (€44.07) diagnoses had higher specialist costs than patients without these diagnoses. Patients living in rural areas have slightly lower costs than patients resident in cities. Furthermore, patients living in more deprived areas have lower costs than those living in areas with lower deprivation.

## Discussion

The cohort analysis reveals that the coordination of care decreased substantially after the abolition of the copayment. Patients with chronic and mental illnesses whose care was previously coordinated by a GP were more likely to consult specialist physicians without referral after the

**Table 2. Results of longitudinal regression modelling.**

| Parameter | Attribute | Effect in € (95%-CI) |
|---|---|---|
| Coordination (reference: CP) | UP | 21.78 (21.46; 22.11) |
| | GP | −102.94 (−103.26; −102.61) |
| | NR | −50.82 (−51.14; −50.49) |
| Copayment (reference: present) | Abolished | −5.55 (−5.94; −5.16) |
| Interaction: Coordination x copayment (reference: CP, copayment present) | UP (copayment abolished) | 2.39 (1.95; 2.83) |
| | GP (copayment abolished) | 1.56 (1.12; 2.00) |
| | NR (copayment abolished) | 0.97 (0.54; 1.40) |
| Time (reference: 1st quarter/2011) | Following quarters | 0.09 (0.07; 0.11) |
| Age group (reference: 18–30) | 31–45 | −0.79 (−1.18; −0.39) |
| | 46–60 | −2.89 (−3.28; −2.51) |
| | 61–75 | −8.59 (−9.03; −8.16) |
| | 76+ | −17.66 (−18.23; −17.10) |
| Sex (reference: male) | Female | 19.36 (18.89; 19.84) |
| Interaction: Age group x sex (reference: 18–30, male) | 31–45 (female) | −10.42 (−10.98; −9.86) |
| | 46–60 (female) | −20.68 (−21.21; −20.15) |
| | 61–75 (female) | −24.36 (−24.92; −23.81) |
| | 76+ (female) | −29.53 (−30.21; −28.84) |
| Settlement Structure (reference: cities) | Towns | −8.36 (−8.65; −8.08) |
| | Rural with densification | −8.93 (−9.17; −8.69) |
| | Sparsely populated | −6.96 (−7.21; −6.71) |
| BIMD 2010 (reference: Q1/lowest deprivation) | Q2 | −3.81 (−4.06; −3.56) |
| | Q3 | −8.00 (−8.26; −7.74) |
| | Q4 | −6.86 (−7.15; −6.56) |
| | Q5 (highest deprivation) | −6.94 (−7.23; −6.65) |
| Psychological disorder (reference: not present) | Present | 22.24 (22.06; 22.42) |
| Chronic illness (reference: not present) | Present | 44.07 (43.85; 44.28) |

BIMD 2010: Bavarian Index of Multiple Deprivation 2010; CP: Coordinated patient; GP: General practitioner care only; NR: Not relevant for coordinated care; UP: Uncoordinated patient; Q: Quintile.

copayment was removed. Coordinated patients had lower specialist costs than similar uncoordinated patients, a previously observed effect [15] that we find to be exacerbated by the abolition of the copayment.

An earlier investigation using the same database indicated that, in 2011, patients with GP coordination had on average €9.65 lower outpatient healthcare costs than uncoordinated patients [15]. In contrast to the present study, the investigation included the cost of prescribed medication and GP costs. We extend understanding of this effect by considering the development of specialist costs among CP and UP patients over an extended period of 6 years, both with and without a copayment for non-referred physician contacts.

The interaction between copayment and abolition indicates that the difference in costs between coordinated (CP) and uncoordinated patients (UP) further increased after abolition. Although the difference between coordinated and uncoordinated patients increased only marginally from €21.78 under the copayment to €24.94 after, it has to be considered that a

substantially smaller proportion were still coordinated by a GP when the copayment was removed.

The evidence on GP-coordinated care and healthcare costs is inconsistent. Investigations by Garrido et al. [35], Delnoij et al. [36] and Starfield [37] showed that systems with gatekeeping or strong primary care have lower healthcare expenditures. In contrast, Kringos et al. found that strong primary care systems are associated with better health, but also with higher expenditures [7]. However, the underlying methods in these investigations do not allow causal inferences.

One explanation lies in the way that primary care is organised, delivered and financed in different nations. Furthermore, the results of Kringos et al. are based on ecological studies [7]. We would expect that the individual patient data of the KVB more precisely capture the relationship between coordination and health care utilization.

The considerable decrease in GP-centred care after 2012 of particular concern. Found in a preceding ecological study based on the same data source and observing the entire Bavarian population [22], our current cohort analysis sheds light on the morbidity structure of patients switching from coordinated to uncoordinated care. After removal of the copayment, the uncoordinated patients exhibited high morbidity, as measured by the proportion with chronic illness and the number of medical condition categories (THCC/RHCC, see also Figure in S1 Fig). Additionally, a higher proportion had a record of psychological disorder, which was previously found to be associated with both uncoordinated specialist contacts and higher costs of ambulatory care [15,20]. This development is to be interpreted more as a change in the composition of the groups than in a change in the morbidity of the individual patients.

The presence of chronic diseases should be considered in a differentiated way. Under the influence of the copayment, the presence of one or more chronic diseases was still a strong predictor for coordinated care [20], but this effect seems to have been weakened by its abolition (see also Figure S1 Fig). One explanation could be the monetary incentive for this collective, as chronic disease patients often have contact with many different specialist groups within a single quarter and €10 was charged for each uncoordinated consultation. After abolition, this sanction no longer existed, and there was no longer any direct monetary incentive to contact a GP before a specialist visit. Consequently, primary care was weakened as a substantially smaller proportion of patients remained coordinated (Figure in S2 Fig). Increasing proportions of uncoordinated care in chronic and mental diseases represent a matter of concern: patients with chronic diseases and multimorbidity would benefit from a strong primary care system [7,38,39], e.g. by reducing mortality due to a higher level of continuity of care [6,10].

## Strengths and limitations

The main strength of our present study is the use of data encompassing all statutorily insured patients in Bavaria, Germany, over a six-year period. For computational reasons, a cohort of 500 000 patients was selected at random from this comprehensive data set. As the underlying database encompass 85% of the Bavarian population, the data can be assumed to be highly representative and generalizable, for example in comparison with the data of individual health insurance data [40,41].

Existing studies investigating the effects of the German copayment were carried out shortly after their introduction [16–18] or immediately before their abolition [15,19,20]. In contrast, our investigation observed two years before and four years after the abolition. We identified one technical report, published in German, which investigated the changes in physician contacts following the abolition [21]. However, it did not investigate the extent of primary care coordination or other structural factors (such as regional differences).

The use of routinely collected claims data in the present study has some limitations, since they were collected initially for billing purposes, not for research. Retrospectively, it is not possible to definitely identify the extent to which a GP referral represents active patient coordination. It is assumed that the proportion of patients who received a referral from a GP overestimates the proportion of patients with active coordination, since patients have the possibility to request a referral to a specialist without a prior appointment with the GP [42]. Conversely, patients for whom no referral was recorded could actually have had some form of GP coordination. This could occur if a patient failed to pass the referral form to the specialist.

Based on the available data, no direct conclusion can be drawn about the quality of care. The outcomes 'coordination status' or 'specialist financial claims' have to be interpreted as surrogate parameters for effective primary care. Following the arguments of Starfield et al. [5] and Forrest & Starfield [43], a decrease in primary care coordination results in a decrease of continuity of care. No mortality or hospitalisation data were available. Consequently, it was not possible to evaluate patient outcomes. Regarding the specialist costs in the case of (un) coordinated care, it should be noted that the estimations depend on the selected model. Choosing a different model would result in different cost amounts. However, sensitivity analyses with different model specifications demonstrate the stability of the cost effect. The combination of modelling and weighting was designed to further improve the robustness of the inference [29–31]. In addition, our models suggest that, notwithstanding an underlying increasing trend, the cost of specialist care decreased slightly (CP: -€5.55; UP: -€3.16) after abolition of the copayment. This is likely caused both by a concomitant change to budgetary process in Bavaria and by efforts to increase the specificity of ICD-10 coding, thus promoting less costly patients to groups that were previously associated with higher costs. Such effects must be viewed as limitations of the data, but do not substantially modify the effect of coordination of care, which is the target of the causal inference.

## Conclusions

Our results indicate that patients incur lower healthcare costs for specialist ambulatory care when their care is coordinated by a general practitioner, regardless of the presence of a copayment for physician consultations. The abolition of the copayment led to a substantial reduction in primary care coordination, particularly affecting patients with mental illness and high morbidity. This is a matter of concern, as coordinated care is especially important in these vulnerable patient groups. Therefore, coordination of care in the ambulatory setting should be strengthened.

## Supporting information

**S1 Checklist. Research checklist.**
(DOCX)

**S1 Fig. Longitudinal regression modelling (generalised estimating equations, GEE), comparing uncoordinated patients (UP) vs. coordinated patients (CP).**
(PDF)

**S2 Fig.** Distribution of probabilities to belong to the coordination subgroups coordinated patients (CP) (a), uncoordinated patients (UP) (b), or GP care only (GP) (c), quarterly.
(PDF)

**S3 Fig. Longitudinal regression modelling (generalised estimating equations, GEE) with (red) and without (blue) weights for coordination (Inverse Probability of Treatment**

**Weighting, IPTW).** Outcome: Ambulatory specialist costs.
(PDF)

**S1 Table. Successive model structure used to determine the probability of coordination.**
Outcome: Coordination status (CP, UP, GP, NR).
(PDF)

**S2 Table. Successive model structure used to estimate ambulatory specialist costs.**
(PDF)

**S3 Table. Characteristics of the cohort at the time of the copayment (2012) and after its abolition (2014), divided into coordination status (CP, UP, GP, NR) (only 1st quarters presented).**
(PDF)

## Author Contributions

**Conceptualization:** Michaela Olm, Ewan Donnachie, Martin Tauscher, Roman Gerlach, Klaus Linde, Werner Maier, Lars Schwettmann, Antonius Schneider.

**Data curation:** Michaela Olm, Ewan Donnachie.

**Formal analysis:** Michaela Olm, Ewan Donnachie, Antonius Schneider.

**Funding acquisition:** Antonius Schneider.

**Investigation:** Michaela Olm, Ewan Donnachie, Antonius Schneider.

**Methodology:** Michaela Olm, Ewan Donnachie, Martin Tauscher, Roman Gerlach, Klaus Linde, Werner Maier, Lars Schwettmann, Antonius Schneider.

**Project administration:** Michaela Olm, Antonius Schneider.

**Resources:** Michaela Olm, Antonius Schneider.

**Software:** Michaela Olm, Ewan Donnachie.

**Supervision:** Michaela Olm, Martin Tauscher, Roman Gerlach, Klaus Linde, Werner Maier, Lars Schwettmann, Antonius Schneider.

**Validation:** Michaela Olm, Ewan Donnachie, Martin Tauscher, Roman Gerlach, Klaus Linde, Werner Maier, Lars Schwettmann, Antonius Schneider.

**Visualization:** Michaela Olm, Ewan Donnachie, Antonius Schneider.

**Writing – original draft:** Michaela Olm, Ewan Donnachie, Antonius Schneider.

**Writing – review & editing:** Michaela Olm, Ewan Donnachie, Martin Tauscher, Roman Gerlach, Klaus Linde, Werner Maier, Lars Schwettmann, Antonius Schneider.

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
