## [Decision Letter · Decision Letter 0]

11 May 2021

PONE-D-21-02946

Ambulatory specialist costs and morbidity of coordinated and uncoordinated patients before and after abolition of copayment: A cohort analysis

PLOS ONE

Dear Dr. Olm,

Thank you for submitting your manuscript to PLOS ONE. After careful consideration, we feel that it has merit but does not fully meet PLOS ONE’s publication criteria as it currently stands. Therefore, we invite you to submit a revised version of the manuscript that addresses the points raised during the review process.

We look forward to receiving your revised manuscript.

Kind regards,

Ferda Halicioglu, Ph.D

Academic Editor

PLOS ONE

Journal Requirements:

4.Thank you for stating the following in the Competing Interests section:

"I have read the journal's policy and the authors of this manuscript have the following competing interests: ED, MT and RG are employees of the Bavarian Association of Statutory Health Insurance Physicians.

The authors MO, KL, WM, LS and AS declare that they have no competing interests."

Reviewers' comments:

Reviewer's Responses to Questions

**Comments to the Author**

1. Is the manuscript technically sound, and do the data support the conclusions?

Reviewer #1: Yes

Reviewer #2: Yes

2. Has the statistical analysis been performed appropriately and rigorously? 

Reviewer #1: Yes

Reviewer #2: Yes

3. Have the authors made all data underlying the findings in their manuscript fully available?

Reviewer #1: Yes

Reviewer #2: No

4. Is the manuscript presented in an intelligible fashion and written in standard English?

Reviewer #1: No

Reviewer #2: Yes

5. Review Comments to the Author

Reviewer #1: The manuscript needs English proofreading.

............................................................................................................................................................................

Reviewer #2: This manuscript deals with an interesting topic. It is well-written and well-structured, and can have a contribution to the international literature on the subject.

The Introduction is concise and informative, while the data and methods are very detailed and transparent.

The results are well-presented and discussed, while the manuscript also includes a concise discussion of the limitations of this work.

Overall, I do not see any major issues with this manuscript.

There is one clarification I would like to see. The authors state that “We performed a retrospective routine data analysis, using claims data of the Bavarian Association of the Statutory Health Insurance Physicians during the period 2011–2012 (with copayment) and 2013–2016 (without copayment), covering 24 quarters.”.

Why are data before 2011 not used? Meaning, why not split the examined period in two equal periods (in terms of number of years)?

Moreover, in the “Cohort size” it is stated that “Between 2012 and 2016, the number of cohort participants decreased by approximately 10%....”. This is a bit confusing and needs rephrasing, as it could be read like the period starts in 2012 and not in 2011.

Finally, I believe that the Figures should be modified in order to increase their appearance. Also, is there any sort of, e.g., heat map or other form of visualization that could be included in the analysis?

6. PLOS authors have the option to publish the peer review history of their article (what does this mean?). If published, this will include your full peer review and any attached files.

Reviewer #1: No

Reviewer #2: No

---

## [Author Response · Author response to Decision Letter 0]

11 Jun 2021

Journal Requirements:

R: Thank you for the style templates. We have adjusted following parts:

- Commas in the author descriptions have been moved to the end, e.g. “Michaela Olm,1¶*” is now “Michaela Olm1¶*,”

- States were added to the author descriptions

- The designations of equal contributorship have been deleted from the title page. The detailed contribution of each author is now listed in the section “Author Contributions”, as we believe that this reflects the level of participation more adequately.

- We added “.tif” to each file naming, e.g. “Fig1” is now “Fig1.tif”

- The format of supporting information citations was changed. “Table S1”, “Table S2”,… or “Fig S1”… are now cited in single files as “S1 Table” , “S2 Table”… or “S1 Fig” and include the designation “.pdf”.

R: We have carefully checked our reference list. All articles are still available and none have been retracted. We noticed that the article by Brenner et al. (2005) can only be found via the German title, although the journal “Zeitschrift für Allgemeinmedizin” provides an English title (but is not indexed in Medline). To enable fast identification of the cited article, the reference has been changed. We now use the German title: “Brenner G, Koch H, Franke A. Steuert die Praxisgebühr in die richtige Richtung? - Analyse des Versorgungsgeschehens nach Einführung der „Praxisgebühr”. Z Allg Med. 2005;81(09):377-81.”

Furthermore, the homepage of the BBSR has received a new launch in June 2020 and the cited link is unfortunately no more available. We have identified the new link to the cited content and updated the reference, from “BBSR. Laufende Raumbeobachtung - Raumabgrenzungen. Siedlungsstrukturelle Kreistypen 2017 [Cited 2020 May 28]. Available from: https://www.bbsr.bund.de/BBSR/DE/Raumbeobachtung/Raumabgrenzungen/deutschland/kreise/Kreistypen4/kreistypen_node.html.” to “BBSR. Raumabgrenzungen: Referenzdateien. [Cited 2021 June 8]. Available from https://www.bbsr.bund.de/BBSR/DE/forschung/raumbeobachtung/downloads/download-referenzen.html”

R: Thank you for your note. We hope to meet your requirements by reformulating our Data Availability statement: “The data that support the findings of this study are held by the Bavarian Association of Statutory Health Insurance Physicians. The data are not publically available due to data protection regulations, but may be made available to researchers upon reasonable request and with the permission of the Bavarian Association of Statutory Health Insurance Physicians (E-mail: versorgungsforschung@kvb.de).”

"I have read the journal's policy and the authors of this manuscript have the following competing interests: ED, MT and RG are employees of the Bavarian Association of Statutory Health Insurance Physicians. The authors MO, KL, WM, LS and AS declare that they have no competing interests."

R: We have updated our Competing Interests section: "I have read the journal's policy and the authors of this manuscript have the following competing interests: ED, MT and RG are employees of the Bavarian Association of Statutory Health Insurance Physicians. This does not alter our adherence to PLOS ONE policies on sharing data and materials. The authors MO, KL, WM, LS and AS declare that they have no competing interests."

Reviewers' comments:

Reviewer's Responses to Questions

Comments to the Author

R: In order to improve readability, questions without reviewer comments (or with answer “Yes”) have been removed (1., 2., and 6.).

3. Have the authors made all data underlying the findings in their manuscript fully available?

Reviewer #1: Yes

Reviewer #2: No

R: For data protection reasons, access to data is restricted. The data holder, the Bavarian Association of Statutory Health Insurance Physicians (BASHIP), is able to provide data access for research purposes. Reasonable requests for data access or additional analysis will be considered by the BASHIP in consultation with its scientific advisory board and data protection officer. In connection with previous studies, the authors have for example provided additional data to assist with interpretation.

We have now reformulated the Data Sharing Statement as follows:

“The data that support the findings of this study are held by the Bavarian Association of Statutory Health Insurance Physicians. The data are not publically available due to data protection regulations, but may be made available to researchers upon reasonable request and with the permission of the Bavarian Association of Statutory Health Insurance Physicians (E-mail: versorgungsforschung@kvb.de). ”

4. Is the manuscript presented in an intelligible fashion and written in standard English?

Reviewer #1: No

Reviewer #2: Yes

5. Review Comments to the Author

Reviewer #1: The manuscript needs English proofreading.

R: One of the authors (ED) is native English speaker. He carefully checked the revised version. Corrections are marked in the manuscript.

Reviewer #2: This manuscript deals with an interesting topic. It is well-written and well-structured, and can have a contribution to the international literature on the subject.

The Introduction is concise and informative, while the data and methods are very detailed and transparent.

The results are well-presented and discussed, while the manuscript also includes a concise discussion of the limitations of this work.

Overall, I do not see any major issues with this manuscript.

R: Thank you very much for your supportive comments.

There is one clarification I would like to see. The authors state that “We performed a retrospective routine data analysis, using claims data of the Bavarian Association of the Statutory Health Insurance Physicians during the period 2011–2012 (with copayment) and 2013–2016 (without copayment), covering 24 quarters.”.

Why are data before 2011 not used? Meaning, why not split the examined period in two equal periods (in terms of number of years)?

R: There were two reasons for starting the observation in the year 2011. First, GP claims data before this point are incomplete for the years 2009 and 2010 because widespread GP-centred contracts were billed via a third-party. After 2011, fewer patients were enrolled in the programmes and the ICD-10 diagnoses of the enrolled patients are available in most cases. Second, breaks in the underlying data make it difficult to perform a comparable analysis before the year 2010.

The following sentence was added to section “Study design”: “Due to limitations in the data quality with respect the present analysis, time periods before 2011 were not considered.” (see lines 102-104)

Moreover, in the “Cohort size” it is stated that “Between 2012 and 2016, the number of cohort participants decreased by approximately 10%....”. This is a bit confusing and needs rephrasing, as it could be read like the period starts in 2012 and not in 2011.

R: We apologize for the confusing formulation. We have corrected the sentence “Between 2011 and 2016,…”. (see line 261)

Finally, I believe that the Figures should be modified in order to increase their appearance. Also, is there any sort of, e.g., heat map or other form of visualization that could be included in the analysis?

R: Thank you for your suggestion. We have added a line in Figures 1-3, which highlights the date of the copayment abolition. The following sentence was added to the figure legends: “vertical line marks the point of abolition”.

We hope that this additional information contributes to a concise presentation. While a heat map can be a very helpful for of communication, we consider that conventional line charts best facilitate simple and accurate comprehension of the time series.

---

## [Editor Report · Decision Letter 1]

16 Jun 2021

Ambulatory specialist costs and morbidity of coordinated and uncoordinated patients before and after abolition of copayment: A cohort analysis

PONE-D-21-02946R1

Dear Dr. Olm,

We’re pleased to inform you that your manuscript has been judged scientifically suitable for publication and will be formally accepted for publication once it meets all outstanding technical requirements.

Kind regards,

Ferda Halicioglu

Academic Editor

PLOS ONE
---

## [Editor Report · Acceptance letter]

18 Jun 2021

PONE-D-21-02946R1 

Ambulatory specialist costs and morbidity of coordinated and uncoordinated patients before and after abolition of copayment: A cohort analysis 

Dear Dr. Olm:

I'm pleased to inform you that your manuscript has been deemed suitable for publication in PLOS ONE. Congratulations! Your manuscript is now with our production department. 

Kind regards, 

on behalf of

Professor Ferda Halicioglu 

Academic Editor

PLOS ONE